# Which programmes and policies across health and community settings will generate the most significant impacts for youth suicide prevention in Australia and the UK? Protocol for a systems modelling and simulation study

Maria Michail ![ORCID],[1] Jo Robinson,[2,3] Katrina Witt,[2,3] Jo-An Occhipinti,[4,5] Adam Skinner,[4] Michelle Lamblin,[2,3] Maria Veresova,[2,3] Dzenana Kartal,[2,3] Justin Waring[6]

For numbered affiliations see end of article.

**Correspondence to**
Dr Maria Michail;
M.Michail@bham.ac.uk

## ABSTRACT

**Introduction** Suicide is a leading cause of mortality among young people aged 15–24 globally. Despite the deployment of comprehensive suicide prevention strategies, we still do not know which interventions, for which groups of young people, for how long and with what intensity could generate the most significant reductions in suicide rates. System dynamics modelling has the potential to address these gaps. SEYMOUR (System Dynamics Modelling for Suicide Prevention) will develop and evaluate a system dynamics model that will indicate which suicide prevention interventions could generate the most significant reductions in rates of suicide and attempted suicide among young people aged 12–25 in Australia and the UK.

**Methods and analysis** A comparative case study design, applying participatory system dynamics modelling in North-West Melbourne (Australia) and Birmingham (UK). A computer simulation model of mental health service pathways and suicidal behaviour among young people in North-West Melbourne will be developed through three workshops with expert stakeholder groups (young people with lived experience, carers, clinicians, policy makers, commissioners). The model will be calibrated and validated using national, state and local datasets (inputs). The simulation model will test a series of interventions identified in the workshops for inclusion. Primary model outputs include suicide deaths, self-harm hospitalisations and self-harm presentations to emergency departments. An implementation strategy for the sustainable embedding of promising suicide prevention interventions will be developed. This will be followed by model customisation, re-parameterisation, and validation in Birmingham and adaptation of the implementation strategy.

**Ethics and dissemination** The project has received approval from the University of Melbourne Human Research Ethics Committee (2022-22885-25971-4), the University of Birmingham Science, Technology, Engineering and Mathematics Ethics Review Committee (ERN_21-02385) and the UK HRA (22/HRA/3826).

## STRENGTHS AND LIMITATIONS OF THIS STUDY

⇒ System dynamics modelling (SDM) is unique in combining existing scientific evidence, empirical data and the experiential knowledge of expert stakeholders to inform model development, evaluation and refinement.
⇒ SDM allows us to simulate the likely impacts of interventions over time on desired outcomes and understand with what intensity and for how long investments are required to sustain the effects of an intervention prior to implementing any changes in the real world.
⇒ Parameter uncertainty and poor data inputs could impact model credibility and usability.

SEYMOUR's dissemination strategy includes open-access academic publications, conference presentations, accessible findings coproduced with young people, e-briefs to policy makers, webinars for service providers and commissioners.

## INTRODUCTION

Suicide is one of the leading causes of mortality among young people aged 15–24 globally.[1] Self-harm and suicidal thoughts and behaviours are complex phenomena linked to a gradual and increasing build-up of vulnerability associated with multiple, interacting, and inter-related biological, psychosocial, cultural, and political factors.[2] The suicide mortality rate is one of the indicators within the UN's Sustainable Development Goal (SDG) 3, and the WHO Comprehensive Mental Health Action Plan (2013–2030) identifies young people as a high-risk group setting out several key strategies for achieving

the global target of reducing the rate of suicide by one-third by 2030. These strategies include improving surveillance data, reducing access to suicide methods, reducing suicide-related stigma, access to medical services and specialist treatment, crisis intervention, and postvention.

Despite the deployment of comprehensive and multi-level suicide prevention strategies,[3] we still do not know which interventions, for which groups of young people, for how long and with what intensity could generate the greatest reductions in youth suicide rates.[4] Therefore, we cannot efficiently and effectively inform decision-making in youth suicide prevention policy and planning.

The success of global and national suicide prevention strategies has been hampered by many factors including:

► A lack of involvement of young people in the design of interventions meaning that they are not youth-focused and, therefore, not acceptable to end-users.[5]
► A lack of planning for the systematic uptake and implementation of strategies in the real world.[6]
► Limited adequacy and accuracy of current methodological approaches (eg, meta-analyses, regression models) for synthesising and operationalising research evidence used to inform suicide prevention strategies.[7] Current methodological approaches do not: (1) respond to real-time suicide data; (2) account for the interdependence of suicide risk factors as they operate across multiple levels (eg, individual, social, health system); and (3) consider the complexity of health systems and the influence of factors such as healthcare constraints, for example, access and capacity.[8 9]

## Potential of system dynamics modelling

Systems thinking and system dynamics modelling (SDM) offer a much-needed paradigm shift in suicide prevention. Systems thinking helps recognise, describe and understand the complex and dynamic interactions within a system, where a system is conceptualised as a complex set of interdependent relationships between heterogenous people, organisations or other units of analysis.[10] SDM is a computer-assisted method that helps frame, test and simulate the causal processes and interactions that underlie complex systems or behaviours to inform policy making.[11] SDM allows policy makers to experiment with different 'what-if' scenarios to understand system behaviour. Most importantly, SDM allows us to simulate the likely impacts of interventions over time on desired outcomes and understand with what intensity and for how long investments are required to sustain the effects of an intervention.

As such, SDM addresses existing gaps in suicide prevention research, policy and practice in three key ways. First, SDM has the capacity to model multiple interacting and inter-related suicide risk factors (*dynamic complexity*) to understand likely trajectories of suicidal behaviour, to model the impacts of alternative interventions and thereby determine what works, for which groups of young people (eg, males vs females) and why. Second, SDM is grounded in the principles of implementation science, thus, maximising opportunities for policy adoption and uptake in the real world.[6 8 12] Third, SDM combines existing scientific evidence, empirical data, and the experiential knowledge of stakeholders (eg, young people, healthcare providers, service commissioners) to coproduce a computational model that simulates what happens in the real world. It is this participatory approach to model development, evaluation and refinement that ensures sufficient implementation planning before rollout.[13–15]

SDM has been previously adopted to inform policy and planning in mental health and suicide prevention. In Australia, SDM has been used as a decision tool to inform both regional[16 17] and national plans for suicide prevention.[18] The adoption, however, of SDM in tackling the increasing suicide rates among young people is still in its infancy[4]; yet its potential to efficiently and effectively guide suicide prevention strategies tailored to the unique needs of young people is great.

## Aims

SEYMOUR (System Dynamics Modelling for Suicide Prevention) will develop and evaluate an SDM to inform youth suicide prevention policy, planning and implementation in Australia and the UK. Specifically, SEYMOUR aims to:

► Develop and validate a model to inform the most appropriate combination of population-level suicide prevention interventions that would generate the most significant reductions in rates of suicide and attempted suicide among young people aged 12–25 over a 10-year period in North-West Melbourne (2023–2033).
► Develop an implementation strategy to facilitate the adoption of the model as a decision-making tool in youth suicide prevention policy and practice in North-West Melbourne.
► Adapt and validate the model in the UK context and optimise how it can inform policy, system-level reform and service redesign in relation to youth suicide.

## METHODS AND ANALYSIS
### Context

SEYMOUR will adopt a comparative case study design and take place in Australia and the UK (February 2022–February 2025). Both countries have experienced a steady increase in suicide rates among young people aged 12–25 over the past 10 years, particularly among young females.[19 20] In 2021, the UK recorded the largest increase in suicide rates among females aged 24 years or under (3.6 deaths per 100 000 females) since 1981 (2.3 deaths per 100 000 females). In Australia, suicide rates for females aged 15–19 and 20–24 increased from 6.1 to 7.1 (per 100 000 females) and from 7.4 to 9.0, respectively, between 2020 and 2021.

In Australia, the project will take place in North-West Melbourne, which includes some of Victoria's most socio-economically disadvantaged and culturally diverse areas.[21]

In the UK, the project will take place in Birmingham—with 40% of its population under 25 and among the most culturally diverse and socioeconomically deprived cities in the UK.

## Methodology and study design

This is a mixed-methods study delivered via three inter-related work packages guided by Forrester's SDM framework[22] which involves problem definition and model conceptualisation, formulation and testing (aim 1); and development of an implementation strategy (aim 2), both of which will take place in North-West Melbourne, Australia. This will be followed by model customisation, re-parameterisation and validation in Birmingham, UK, and adaptation of the implementation strategy (aim 3).

The study is based on a participatory modelling approach which allows for a diverse group of stakeholders to be actively involved in conceptually mapping causal pathways for the development of suicidal behaviour in young people and mental health service pathways in a region.[15] Stakeholders can provide unique knowledge on how the mental health system operates in their region; identify weaknesses in the system (ie, leverage points); suggest and evaluate ideas for strategic interventions, policies or service planning decisions that would optimise the structure of the system and how it works (ie, mechanisms of effect). Importantly, stakeholder knowledge of the local barriers and facilitators of translation of different interventions can inform the development of implementation strategies tailored to the local context.[13] The unique combination of expert knowledge and empirical data has the potential to enhance model credibility and utility,[23] as well as the translation of the model into effective suicide prevention policy and practice in Australia and the UK.

## Sampling strategy

This sampling strategy applies across all work packages. Purposive sampling will be used to ensure maximum variation in the perspectives, cases and expertise recruited, including:

► Young people aged 12–25 years with lived/living experience of self-harm and/or suicidal behaviour (regardless of diagnosis), living within the catchment area of North-West Melbourne/Birmingham.
► Staff from non-governmental organisations (NGOs) who provide services to young people with lived/living experience of self-harm and/or suicidal behaviour, operating within the catchment area of North-West Melbourne/Birmingham.
► Family members/carers of young people aged 12–25 with lived experience of self-harm and/or suicidal behaviour, living within the catchment area of North-West Melbourne/Birmingham.
► Primary and community healthcare practitioners including general practitioners, practice nurses, community nurses, nurse practitioners, allied health and mental health professionals working within

the catchment area of North-West Melbourne/Birmingham.
► Regional suicide prevention policy leads, service planners, health and social care policy makers, operating within state of Victoria/West Midlands.

## Recruitment procedures

Two expert stakeholder groups (Australia and UK) will be recruited consisting of 10–15 participants from the aforementioned groups.

### Australia

Young people aged 12–25 will be recruited from the five headspace centres (headspace centres in Australia provide multidisciplinary frontline care to young people aged 12–25) operated by Orygen (Orygen is an internationally acknowledged youth mental organisation with expertise in research, policy, education and innovations in care for young people) and serving the catchment area of North-West Melbourne: Werribee, Glenroy, Craigieburn, Sunshine and Melton. The research study team will liaise with clinical staff to determine participant eligibility. For young people under the age of 18, consent will be sought from parents/legal guardians.

Family members or carers of young people aged 12–25 with lived/living experience of self-harm and/or suicidal behaviour will be recruited from the same five headspace centres and local parent support groups.

Primary and community healthcare practitioners will be recruited via North-West Melbourne Primary Health Network (PHN) (PHNs are independent organisations funded by the Australian Government to coordinate primary healthcare in their respective region) health services. The study poster will be uploaded on the North-West PHN website and practitioners interested in being involved will contact the research team directly. Participants will also be recruited via the five headspace centres and other professional networks.

Regional suicide prevention policy leads, service planners and healthcare policy makers will be recruited across a range of professional networks and governmental organisations such as the Department of Health and Human Services, Victoria.

### UK

Young people aged 12–25 as well as family members/carers will be recruited using social media and from community settings, youth clubs and parent support groups across Birmingham. Primary, secondary and community healthcare practitioners will be recruited via three National Health Service (NHS) sites: Birmingham and Solihull Clinical Commissioning Group, the largest clinically led commissioning organisation in England, which includes 170 general practitioner practices; Birmingham Community Healthcare NHS Foundation Trust, where we will recruit a range of health professionals including community paediatricians, school nurses and nursing staff; Birmingham Women's and Children's

NHS Foundation Trust, which hosts Forward Thinking Birmingham, the city's youth mental health service for 0–25 years old. NGOs staff will be recruited via social media. Regional suicide prevention policy leads, service planners and policy makers will be recruited via professional networks and governmental organisations such as the Birmingham Suicide Prevention Steering Group and the West Midlands Combined Authority.

### Procedure

#### Work package 1: development and validation of a youth suicide prevention SDM in North-West Melbourne, Australia

A computer simulation model of mental health service pathways and suicidal behaviour among young people will be developed through three, 1-day participatory workshops with the expert stakeholder group. The workshops will take place at a convenient location (eg, community hall) and will be facilitated by members of the research team (MM and KW). The facilitators have significant expertise in workshop facilitation and have received the Applied Suicide Intervention Skills Training. Participants will be reimbursed for their time and travel.

Work package will consist of the following phases

#### *Phase 1: model conceptualisation, formulation, calibration and validation*

##### Model conceptualisation

Two workshops will be conducted with the expert stakeholder group. At workshop 1, the research team will introduce the basic principles of systems modelling to the participants to ensure they can meaningfully engage in the model development process. The expert stakeholder group will then conceptualise and map the pathways for the development of self-harm and suicidal behaviour in young people identifying risk and protective factors and mapping community and mental health service pathways and the factors that influence the flow of the population along these pathways. The expert stakeholder group will work together to prioritise the interventions, policy initiatives or programmes that they would like to see included in the model to capture their effects. This exercise will take place as a facilitated discussion and voting in line with Freebairn *et al*.[15] By the end of the first workshop, the expert stakeholder group will have codesigned a draft conceptual map of the youth mental health system structure in North-West Melbourne.

At workshop 2, the research team will present the draft model structure and logic to the expert stakeholder group for review and refinement. An intervention mapping exercise will follow where participants will define and describe the prioritised interventions (eg, components, mechanisms of effect, unintended consequences, scalability) and identify where each intervention is likely to have its effect in the model structure (*dynamic hypotheses*). The identification and prioritisation of relevant interventions for modelling will be based on (1) current evidence-base (ie, empirical, peer-reviewed research identified by the research team, eg, Cochrane systematic reviews)

which will be presented to the participants during the workshops; and (2) local site needs. This could include suicide-specific interventions, public health mental health interventions, mental health service planning interventions (eg, programmes targeting access, capacity, resources, workforce) both already implemented in the real world and hypothetical interventions as both will be useful for *what-if* scenario testing.[15]

Workshops will be audio recorded and the research team will keep summary points (not ascribed to specific participants) using field notes, post-it notes, butcher's paper, whiteboard. Field notes will be analysed using codebook thematic analysis.[24]

##### Formulation and model calibration

Between workshop 1 and 2, the research team will use the conceptual map to create a computational simulation model using Stella Architect (version 3.1.1). The structure and parameterisation of the model will draw on national, state and local datasets (inputs), for example, prevalence of mental health disorder and help-seeking data (Australian Bureau of Statistics National Survey of Mental Health and Wellbeing, 2020–2021); prevalence of distress (Victorian Population Health Survey); demographic data (Australian Bureau of Statistics); community mental health services (North-West Melbourne PHN). Primary model outputs will include suicide deaths, self-harm hospitalisations (used as proxy for suicide attempts where such data is not captured[4]) and self-harm presentations to emergency departments (EDs) (Victoria Emergency Minimum Dataset). Estimates of the effects of the prioritised interventions and the mechanisms of action will be based on empirical, peer-reviewed research identified by the research team (eg, Cochrane systematic reviews). Parameter values that cannot be derived from available data will be estimated via constrained optimisation. Powell's method[25] will be employed to obtain the set of (optimal) parameter values minimising the sum of the mean absolute per cent error calculated for each time series separately (ie, the mean of the absolute differences between the observed time series values and the corresponding model outputs, where each difference is expressed as a percentage of the observed value). The research team will also engage with key stakeholders (eg, Victoria Department of Health) for expert advice and direction to relevant data sources that would facilitate model calibration and validation.

##### Model verification and validation

Verification and validation of the simulation model will be carried out during phase 1 to ensure the model is acceptable for its intended purpose and accurate. The model will be validated by (1) testing whether the model could replicate historic time series data across a range of key indicators (eg, self-harm hospitalisations, ED presentations, suicide deaths). Outputs will be calculated every 0.875 days (the numerical integration time step, dt, will be set to one-eighth of a week[26] over a period of 20 years,

starting from 2013, permitting comparisons of model outputs with historic data from 2013 to 2023, and forecasts of the impacts of intervention/policy scenarios simulated from the time of implementation (2023) to 2033; (2) ensuring face validity of the model structure and performance by consulting our expert stakeholder group about whether the model and/or its behaviour are reasonable. During the participatory model building process, stakeholders will interact directly with the model interface to provide feedback on its design and functionality, run scenarios, test alternative assumptions, discuss results.[27]

### Phase 2: user interaction with the model, policy testing and sensitivity analysis

The third workshop will be an opportunity for the research team to demonstrate the final version of the systems model structure and logic. The simulation model will incorporate a series of interventions identified in the workshops for inclusion for example, specific suicide prevention interventions, public mental health interventions and mental health service interventions. The expert stakeholder group will interact with the tool to run a set of interventions or policy initiatives to understand the effects generated by the model as well as trade-offs between different scenarios. The research team will interpret the results of the simulated scenarios and draw out initial model insights with regards to what interventions (or combination of interventions) would generate the most significant reductions in rates of suicide, self-harm hospitalisations and self-harm presentations to EDs among young people aged 12–25 over a 10-year period (2023–2033) in North-West Melbourne. Intervention scenarios will be compared against a baseline (business as usual), in which existing programmes or policies remain in place until the end of the simulation. Sensitivity analyses will be performed to assess the impact of uncertainty in estimates of the direct effects of each intervention on the simulation results. Latin hypercube sampling will be used to draw 100 sets of values for selected model parameters determining the direct effects of the interventions on cases of hospital-treated attempted suicide and cases of suicide from a uniform joint distribution spanning ±20% of the default values. Differences in projected outcomes between the baseline and intervention scenarios will be calculated for each set of parameter values and summarised using simple descriptive statistics.[4]

### Work package 2: development of implementation strategy of the SDM

This work package will provide evidence on the optimal strategy for the implementation and sustainable embedding of the selected suicide prevention interventions for young people in North-West Melbourne based on the modelling.

### Phase 1: contextual assessment of implementation barriers and facilitators

Phase 1 will identify the contextual and procedural factors that influence the translation and adoption of the interventions in North-West Melbourne. We will conduct qualitative interviews and focus groups with different stakeholder groups to explore a wide range of views about the benefits, barriers to and consequences of implementing the SDM in North-West Melbourne. The topic guide (online supplemental material 1) was informed by an existing participatory system modelling evaluation framework[28] and the Consolidated Framework for Intervention Research (CFIR[29 30]) interview guide tool to ensure a systematic and comprehensive assessment of potential barriers and facilitators of implementation of the SDM (eg, barriers relevant to the intervention itself, barriers relevant to the inner setting, outer setting).

#### Participants

Participants from work package 1 will be invited to take part in work package 2. Work package 2 will also be open to new participants. We will follow the same sampling and recruitment strategy as per the Methods and analysis section.

#### Data collection and analysis

We will conduct semi-structured qualitative interviews with (1) 20 young people with lived/living experience of self-harm and/or suicidal behaviour; (2) 12 suicide prevention policy leads and service commissioners and (3) 12 health and social care ministerial advisors and policy makers across Victoria.

We will conduct three focus groups (6–8 participants per focus group) with healthcare practitioners and allied health professionals across primary and community-based services across North-West Melbourne; and three focus groups with staff from local/regional NGOs.

Focus groups and interviews will be audio-recorded, transcribed and edited to remove names to preserve participant anonymity. MAXQDA 2022[29] will be used for data coding and analysis. Data will be analysed using framework analysis,[30] a structured and rigorous process of theme-based analysis through the development of charts. Adopting framework analysis will allow the research team to work independently and then collaboratively to develop, review and refine themes and subthemes through an iterative process. This will allow the team to identify a wide range of views about the benefits, barriers to and consequences of implementing the SDM within and across the different stakeholder groups. The CFIR[31 32] will be adopted as a theoretical framework to help us conceptualise the findings and guide the context-specific assessment of the barriers and facilitators.

### Phase 2: a context-specific implementation plan for the SDM

Phase 2 will be informed by the contextual assessment of barriers and facilitators of phase 1 and will develop an implementation strategy for the sustainable embedding

of the selected interventions/policies in North-West Melbourne based on the modelling. We will carry out two workshops with participants from phase 1 to identify and tailor implementation strategies to mitigate barriers and leverage facilitators. The Expert Recommendations for Implementing Change (ERIC[33]) will be used to draw up a list of potential implementation strategies; and the CFIR-ERIC Matching Tool[34] will be used to facilitate the accurate choice of implementation strategies that could help address the phase 1 CFIR-based barriers.

### Participants

Ten to fifteen stakeholders drawn from phase 1 will be recruited using purposive sampling to ensure maximum variation for example, young people with lived/living experience of self-harm and/or suicidal behaviour; family/carers; health and social care policy makers; NGOs and primary healthcare providers.

### Data collection and analysis

At workshop 1, the research team will present the findings of phase 1 in relation to the barriers and facilitators for the implementation success of the suicide prevention interventions in our model. Participants will complete an online questionnaire, using Qualtrics, which will ask them to decide on the most important implementation barriers and associated strategies to address each chosen barrier (online supplemental material 2). Data will be analysed using IBM SPSS Statistics (v.27).[35] Counts and percentages will be used to report data on endorsement of implementation strategies. Those strategies with the highest cumulative endorsement will be selected to be included in the implementation strategy. Descriptive statistics will be used to present data on the most commonly reported variables influencing participants' rankings of the implementation strategies. Comments provided by participants in the free-text boxes will be coded by themes, which will be inductively derived based on descriptive coding of content.[36]

At workshop 2, the research team will present the findings of the survey to all participants and through discussion the group will work together to agree on: (1) a strategy that would guide the implementation of the suicide prevention interventions in practice across North Western Melbourne; and (2) a 'change map', that is, a graphical depiction of the pathway to long-term implementation of the SDM that can be adapted and transferred to other geographical, cultural and service contexts. By the end of work package 2, we will have developed an implementation strategy to facilitate the sustainable embedding of promising suicide prevention interventions for young people in North-West Melbourne.

### Work package 3: UK model customisation, re-parameterisation, validation and adaptation of implementation strategy

In this work package, we will translate and validate the computational model and implementation strategy developed in Australia into the UK context. This work package will be delivered in two phases.

### *Phase 1: adaptation and validation of the Australian conceptual model*

We will carry out two workshops with the UK expert stakeholder group (see the Methods and analysis section). At workshop 1, the research team will present the model originally developed in Australia and through discussion and debate, the conceptual model will be revised to reflect pathways and service provision relevant to the Birmingham context. An intervention mapping exercise similar to the one in work package 1 will be carried out to prioritise interventions for inclusion in the model and identify where each intervention is likely to have its effect in the model structure.

The research team will subsequently create a mathematical representation of the adapted conceptual model. The structure and parameterisation of the model will draw on secondary data sources (inputs) for example, demographic data (Office of National Statistics); data on mental health hospitalisations, suicide attempts/deaths in Birmingham (Clinical Record Interactive Search system; Hospital Episode Statistics); and data on prevalence of mental disorder, and service capacity (Children and Young People's Health Services Data Set). The computer simulation model will run a series of population-level interventions prioritised in the workshops for inclusion to identify which combination of these interventions generates the most significant reductions in rates of suicide and attempted suicide among young people (stratified by males/females) during a 10-year period (2025–2035). Estimates of the effects of these interventions, and the mechanisms of action, will be based on empirical, peer-reviewed research identified by the team (eg, Cochrane reviews) and informed by stakeholder feedback.

At workshop 2, the research team will demonstrate to the group the adapted model structure and logic. User interaction with the model and testing of different intervention scenarios will take place as per work package 1.

### *Phase 2: adaptation of the implementation strategy for the Birmingham context*

This work package will provide evidence on the optimal strategy for implementation of the SDM to inform policy making and service redesign in relation to youth suicide prevention in Birmingham. We will hold two workshops and an online survey with the expert stakeholder group.

Workshop 1 will involve presenting the findings from phase 1 to the expert stakeholder group and sharing the implementation blueprint developed in Australia. We will explore the group's views on the barriers and facilitators for the implementation success of the adapted SDM using the CFIR interview guide tool.[31] Participants will then be asked to complete an adapted version of the online survey questionnaire used in work package 2 (phase 2) where they will be presented with a range of implementation strategies (using the CFIR-ERIC Matching Tool) and asked to rank those strategies they perceive to be more relevant and responsive to the local context.

At workshop 2, the research team will present the survey findings to the expert stakeholder group and through discussion the group will adapt and revise the implementation strategy that would guide the sustainable embedding of the promising suicide prevention interventions for young people in Birmingham.

## Patient and public involvement

The conception and development of the study have been informed by input received by young people during consultations with the Institute of Mental Health Youth Advisory Group, University of Birmingham, UK. Young people advised on the participatory study design (workshops); ethical considerations and safety issues. Young people from Orygen's Youth Council have informed the dissemination and public engagement strategy.

## ETHICS AND DISSEMINATION

The project has received ethics approval by The University of Melbourne Human Research Ethics Committee (2022-22885-25971-4), The University of Birmingham Science, Technology, Engineering and Mathematics Ethics Review Committee (ERN_21-02385) and the UK Health Research Authority (22/HRA/3826). The process of obtaining informed consent will be in accordance with all applicable regulatory requirements in each country including those required for consenting participants under the age of 16 in the UK and 18 in Australia. The main psychological risk is that young people may become distressed during the workshops or may disclose current suicide risk. SEYMOUR's Risk Management Strategy, informed by young people from youth advisory groups, is included in online supplemental material 3.

SEYMOUR's dissemination strategy includes academic publications in open access scientific journals; presentations at international conferences; social media platforms; accessible findings (infographics, podcasts) coproduced with young people; e-briefs to policy makers and ministers, round table discussions for service providers and commissioners.

## DISCUSSION

SEYMOUR offers a novel paradigm for guiding the efficient and effective deployment of national and global suicide prevention strategies. The application of SDM to suicide prevention policy, planning and implementation is still in its infancy.[37] SEYMOUR has the potential to help us unpack the 'black box' in suicide prevention; that is, understand the dynamic complexity of the aetiological factors underlying suicidal behaviour in young people within a complex health and social care services system. SEYMOUR will bridge the gap between methodological approaches, such as participatory action research and simulation modelling, to help us address some of the most pressing and, so far, unsolved questions in suicide prevention: What interventions can effectively and efficiently reduce the increasing rates of suicide and attempted suicide among young people? In doing so, this programme of work has the potential to facilitate the transferability, internationalisation and adoption of SDM as a decision-making tool in youth suicide prevention policy and practice globally to meet the mental health needs of young people and help reduce suicide and its associated burden.

## Author affiliations

[1]School of Psychology, Institute for Mental Health, University of Birmingham, Birmingham, UK
[2]Orygen The National Centre of Excellence in Youth Mental Health, Parkville, Victoria, Australia
[3]Centre for Youth Mental Health, The University of Melbourne, Melbourne, Victoria, Australia
[4]Brain and Mind Centre, Faculty of Medicine and Health, The University of Sydney, Sydney, New South Wales, Australia
[5]Computer Simulation & Advanced Research Technologies (CSART), Sydney, New South Wales, Australia
[6]School of Social Policy, Health Services Management Centre, University of Birmingham, Birmingham, UK

**Contributors** MM, JR, JW conceived the study idea. MM, JR, JW, JO designed the study. AS and KW provided methodological input and advice on data analysis. KW, DK, MV, ML will support data collection, analysis and implementation of the study. MM drafted the manuscript and all authors contributed to the edit of the manuscript. All authors read and approved the final version.

**Funding** This work is supported by the European Union H2020-EU.1.3.2. MSCA-IF-GF—Global Fellowships (grant agreement ID: 101026065) awarded to MM. JW is funded by the European Union H2020-EU.1.3.2. MSCA-IF-GF—Global Fellowships (grant agreement ID: 101026065). JR is funded by a National Health and Medical Research Council Investigator Grant (GNT2008460) and a Dame Kate Campbell Fellowship from the Faculty of Medicine, Dentistry and Health Sciences at the University of Melbourne. KW is funded by a National Health and Medical Research Council Investigator Grant (GNT1177787) and a Dame Kate Campbell Fellowship from the Faculty of Medicine, Dentistry and Health Sciences at the University of Melbourne. AS is supported by philanthropic funding from the Grace Fellowship. JO is funded by the National Health and Medical Research Council Centre of Research Excellence 'Reducing suicidal thoughts and behaviours in young people presenting for health care' (1171910).

**Competing interests** JO is both Head of Systems Modelling, Simulation & Data Science at the University of Sydney's Brain and Mind Centre and Managing Director of Computer Simulation & Advanced Research Technologies (CSART).

**Patient and public involvement** Patients and/or the public were involved in the design, or conduct, or reporting, or dissemination plans of this research. Refer to the Methods section for further details.

**Patient consent for publication** Not applicable.

**Provenance and peer review** Not commissioned; externally peer reviewed.

**ORCID iD**
Maria Michail http://orcid.org/0000-0001-7380-3494

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
