## [Reviewer comments · BMJ Open]

ARTICLE DETAILS

TITLE (PROVISIONAL)	Which programs and policies across health and community settings will generate the most significant impacts for youth suicide prevention in Australia and the UK? Protocol for a systems modelling and simulation study.
AUTHORS	Michail, Maria; Robinson, Jo; Witt, Katrina; Occhipinti, Jo-an; Skinner, Adam; Lamblin, Michelle; Veresova, Maria; Kartal, Dzenana; Waring, Justin

VERSION 1 – REVIEW

REVIEWER	Giabbanelli, Philippe J. Miami University
REVIEW RETURNED	07-Feb-2023

GENERAL COMMENTS	Suicide among young people is a pressing concern and I welcome the authors' proposed study protocol as it would provide useful information to understand the complex system of youth suicide, identify interventions, and assist towards their implementation. As this is a protocol, 'results' are not applicable and the focus is on the adequacy of the proposed methods given existing works and the intended objective. Overall, I do not see a need to change any of the proposed methods. However, I have several clarification questions pertaining to justifying the methods, explaining them, or stating their limitations. I look forward to reading the revised version of this protocol. 1) The sampling strategy aims to cover five groups of participants, for a total of ten to fifteen participants. There is considerable heterogeneity within each group. For example, young people with a history of self-harm or suicidal behaviour are extremely diverse. There are specific risks for minorities (e.g., LGBTQ), different patterns of suicide attempts and death by gender, etc. See for example "Intersectionality and health behaviors among US high school students: examining race/ethnicity, sexual identity, and sex" or "Correlates of past year suicidal thoughts among sexual and gender minority young adults". I wonder whether an upper-average of $15/5 = 3$ participants from each category would really suffice to deal with in-group heterogeneity. The risk would be that the model only represents 'one majority', which has lately been a criticism of some suicide prevention strategies. 2) I appreciate that the workshops will provide input on interventions. However, I believe that both the UK and Australia already have intervention packages, which often cover both national-level policies and their local implementations. For example, in the UK there is the "Suicide prevention in England: fifth progress report", the "Suicide prevention: cross-government plan", and most recently research briefing cbp-8221 by the UK
--

	Parliament. Given the immense efforts that went into identifying interventions both nationally and locally, the workshop seems to be starting from scratch with a smaller (albeit adequately diverse) panel. I am concerned that if interventions would only be identified through one workshop, it would not be as comprehensive as the current evidence base. I thus wonder whether (i) the evidence base could be shared with workshop participants or (ii) there is a point that I missed as to why interventions would stem solely from the workshop instead of accounting for existing packages. 3) Participatory modeling can operate via causal maps and then System Dynamics, as proposed here. It's a classical approach, well-tested, with a proven ability to provide time estimates as long as there is sufficient input data. However, perhaps because of its brevity, the protocol seems to only be covering the merits of SD without accounting for methods that are closely related. "Tools and methods in participatory modeling: Selecting the right tool for the job" has a comprehensive overview of techniques and can be used to justify the methodological choices made here; "Comparing, Choosing, and Combining Systems Mapping Methods" is also a good starting point. The most closely related technique of Fuzzy Cognitive Maps can combine scientific evidence for public health policymaking, using empirical data together with stakeholders' qualitative views (see "Dealing with complexity: How to use a hybrid approach to incorporate complexity in health behavior interventions"). 4) Several systems maps have been created for suicide, and particularly youth suicide. Recent examples include "Suicide attempts during adolescence: Testing the system dynamics of the interpersonal theory of suicide" and "Pathways to suicide or collections of vicious cycles? Understanding the complexity of suicide through causal mapping". The interdependence of youth suicide risk factors across levels and domains has thus been well studied in several places. In light of this evidence base, I would like more clarification on the research gap that would be addressed by conducting workshops to create another map of suicide. Perhaps the gap is the location (Australia / UK) or the perspectives (most existing maps are expert-centered and this proposal has a unique angle in involving young people and family members) -- it would be best for the authors to state the gap that they intend to address. 5) The description of the workshops is extremely short ("The expert stakeholder group will then conceptualise and map the pathways"). There are many ways to run these workshops (see "Running Systems Mapping Workshops" by Barbrook-Johnson). For example, is there a trained facilitator, and what are their roles? How long is the workshop? How do you account for power imbalances? How do you make it a safe space, particularly for youth with a history of self-harm and their family? What about barriers to access (e.g., getting to the workshop location) and compensation mechanisms, to avoid self-selection on criteria such as being able to own a car or having flexible work hours? 6) Reading that "The expert stakeholder group will work together to prioritize the interventions, policy initiatives or programs" seems like a lot of work for one workshop. That may require specialized tools in order to deal with the system. For example, if the workshop results in a set of sticky notes connected by hand-drawn lines on a whiteboard, participants may be looking at a spaghetti plate and struggle with designing interventions. How will participants be helped in navigating the suicide map in order to look for interventions?
--	---

	7) The parametrization of an SD model for suicide is a project by itself. There are plenty of unknowns in suicide (particularly at the community and societal levels), and navigating a map of suicide given these data limitations is tricky (see "The necessity and difficulty of navigating uncertainty to develop an individual-level computational model"). How are the authors planning to deal with the unknowns when operationalizing the model? 'Constrained optimization' is nice but if there is a lot of unknowns, there is a very large search space which brings both computational models and may lead to identifying values that are not actually possible. 8) The workshops are intended to look at the complex situations surrounding people's lives in a local setting. However, every dataset mentioned in this project is at the national level. There seems to be a disconnect between the local workshops and the datasets available. Can the authors provide a table about which datasets would be used and for which domain (e.g., local healthcare resources, family dynamics, substance use and abuse...)? 9) Verification and validation are extremely important in a model, and they are two distinct activities. I could not find how verification would be carried out in the descriptions. As for validation, we expect the discrepancy between simulated outcomes and expected outcomes to be within an acceptable range, for example. It is fine to have some 'extreme testing' by running a few scenarios and asking stakeholders that the projections are 'plausible', but it does not replace the need for comparisons of point estimates. Can the authors clarify what their model's outputs will be compared against, and what would be an acceptable accuracy? 10) The proposal is interesting as it deals with resources, but that aspect seems to be missing from the process. For example, to find which combination of interventions have the most significant reductions, the answer could just be: all of them. But that is not a very useful answer, because we do not have the resources to perform every single intervention. Healthcare constraints or a cost-effectiveness aspects are important to guide intervention design and planning, otherwise the model can only tell us about the expected effect of an intervention but cannot be used to prioritize them. On the design of simulation tools for youth suicide prevention, it may be useful to look at "Design and Deployment of a Simulation Platform: Case Study of an Agent-Based Model for Youth Suicide Prevention". 11) Translating from Australia to the UK is a very interesting final work package. Reading that the "conceptual model will be revised to reflect pathways and service provision relevant", I wonder whether this has ever been done. As in, is there prior work on creating a systems map with one group of participants and then asking another group to adapt it? My concern here is about confirmation bias. If I show a suicide map to a set of stakeholders, there would be a risk that they endorse most of its content, but they would not have produced it themselves if we ran the workshop from scratch. 12) I am glad to see that "young people advised on the participatory study design (workshops); ethical considerations and safety issues." However, I believe that sharing these details would be important for the protocol. There are definitely risks inherent to having conversations with young people who have a history of self-harm behaviors, and the management of these risks throughout the workshop is critical.
--	--

VERSION 1 – AUTHOR RESPONSE

Reviewer 1

1) The sampling strategy aims to cover five groups of participants, for a total of ten to fifteen participants. There is considerable heterogeneity within each group. For example, young people with a history of self-harm or suicidal behaviour are extremely diverse. There are specific risks for minorities (e.g., LGBTQ), different patterns of suicide attempts and death by gender, etc. See for example "Intersectionality and health behaviors among US high school students: examining race/ethnicity, sexual identity, and sex" or "Correlates of past year suicidal thoughts among sexual and gender minority young adults". I wonder whether an upper-average of $15/5 = 3$ participants from each category would really suffice to deal with in-group heterogeneity. The risk would be that the model only represents 'one majority', which has lately been a criticism of some suicide prevention strategies.

Response: We would like to thank the reviewer for raising this important issue. The sample size of participatory model development workshops can vary significantly depending on project needs, feasibility, timelines, and resources (Freebairn et al, 2016; Occhipinti et al, 2021; Giabbanelli et al, 2022). There are examples of studies using participatory methods in modelling with fewer than 10 participants (van Vliet et al, 2010). Whilst the aim is to achieve phenomenal and demographic variation in our sample, the proposed number of participants (10 to 15) allows for wide yet meaningful engagement with key stakeholders, ensuring that conversations are carried out in a respectful and manageable way (Freebairn et al, 2016). In line previous studies in the field of youth mental health and/or suicide prevention (Freebairn et al, 2022), our priority is to meaningfully embed local and contextual knowledge into the participant recruitment process. In addition, we do intent on doing any subgroup analysis, rather we are aiming to achieve saturation of themes across the group as a whole.

2) I appreciate that the workshops will provide input on interventions. However, I believe that both the UK and Australia already have intervention packages, which often cover both national-level policies and their local implementations. For example, in the UK there is the "Suicide prevention in England: fifth progress report", the "Suicide prevention: cross-government plan", and most recently research briefing cbp-8221 by the UK Parliament. Given the immense efforts that went into identifying interventions both nationally and locally, the workshop seems to be starting from scratch with a smaller (albeit adequately diverse) panel. I am concerned that if interventions would only be identified through one workshop, it would not be as comprehensive as the current evidence base. I thus wonder whether (i) the evidence base could be shared with workshop participants or (ii) there is a point that I missed as to why interventions would stem solely from the workshop instead of accounting for existing packages.

Response: We thank the reviewer for the opportunity to clarify this point. The identification and prioritisation of relevant interventions for modelling will be based on i) current evidence-base (i.e., empirical, peer-reviewed research identified by the research team e.g., Cochrane systematic reviews) which will be presented to the participants during the workshops; and ii) local site needs. This will include both interventions already implemented in the real world and hypothetical interventions as both will be useful for what-if scenario testing. We have now revised the relevant section in the manuscript to reflect this (page 7).

3) Participatory modelling can operate via causal maps and then System Dynamics, as proposed here. It's a classical approach, well-tested, with a proven ability to provide time estimates as long as there is sufficient input data. However, perhaps because of its brevity, the protocol seems to only be covering the merits of SD without accounting for methods that are closely related. "Tools and methods in participatory modelling: Selecting the right tool for the job" has a comprehensive overview of techniques and can be used to justify the methodological choices made here; "Comparing, Choosing,

and Combining Systems Mapping Methods" is also a good starting point. The most closely related technique of Fuzzy Cognitive Maps can combine scientific evidence for public health policymaking, using empirical data together with stakeholders' qualitative views (see "Dealing with complexity: How to use a hybrid approach to incorporate complexity in health behavior interventions").

Response: There are indeed various tools and methods to support participatory modelling (Voinov et al, 2016) but reviewing the pros and cons of each tool/method to justify the use of SD for this study is beyond the scope of this paper. SD was most appropriate for the purposes of this study based on the research question, project needs, model outputs and the what-if scenario analyses and predictions we will be testing (Voinov et al, 2016).

4) Several systems maps have been created for suicide, and particularly youth suicide. Recent examples include "Suicide attempts during adolescence: Testing the system dynamics of the interpersonal theory of suicide" and "Pathways to suicide or collections of vicious cycles? Understanding the complexity of suicide through causal mapping". The interdependence of youth suicide risk factors across levels and domains has thus been well studied in several places. In light of this evidence base, I would like more clarification on the research gap that would be addressed by conducting workshops to create another map of suicide. Perhaps the gap is the location (Australia / UK) or the perspectives (most existing maps are expert-centered and this proposal has a unique angle in involving young people and family members) -- it would be best for the authors to state the gap that they intend to address.

Response: The purpose of the specific mapping exercise is to understand likely trajectories of suicidal behaviour of young people in North-West Melbourne and Birmingham whilst considering the complexity of health systems in the respective regions and the factors -within those regions- that influence the flow of the local population along these pathways such as capacity, access, and workforce. Therefore, the systems maps that will be developed will be regional and reflect how local systems operate, weaknesses in the systems that could be used as leverage points, and how the systems could be optimized through different policies, interventions, or service planning decisions relevant to the needs of the specific regions.

5) The description of the workshops is extremely short ("The expert stakeholder group will then conceptualise and map the pathways"). There are many ways to run these workshops (see "Running Systems Mapping Workshops" by Barbrook-Johnson). For example, is there a trained facilitator, and what are their roles? How long is the workshop? How do you account for power imbalances? How do you make it a safe space, particularly for youth with a history of self-harm and their family? What about barriers to access (e.g., getting to the workshop location) and compensation mechanisms, to avoid self-selection on criteria such as being able to own a car or having flexible work hours?

Response: We have now provided further information about the workshops (pages 6, 11).

6) Reading that "The expert stakeholder group will work together to prioritize the interventions, policy initiatives or programs" seems like a lot of work for one workshop. That may require specialized tools in order to deal with the system. For example, if the workshop results in a set of sticky notes connected by hand-drawn lines on a whiteboard, participants may be looking at a spaghetti plate and struggle with designing interventions. How will participants be helped in navigating the suicide map in order to look for interventions?

Response: During the development of the conceptual map, the facilitators will work with the expert stakeholder group to identify weaknesses in the system (i.e., leverage points). The identification and prioritisation of interventions or policies, aiming to optimise the structure of the system and how it works, will run as a facilitated discussion and voting exercise in line with Freebairn et al (2022). The

research team will present a slide showing a list of potential interventions based on the current evidence-base. Stakeholders will be requested to suggest additional/alternative interventions (not included in the list) based on local knowledge and needs by using sticky notes and placing these on a wall surface. The research team will present a slide with the combined interventions grouped based on themes e.g., suicide-specific interventions, mental health service delivery interventions. The interventions grouped by themes will also be presented on butcher's paper on a wall surface. Participants will be given sticky dots and asked to choose the interventions they would like to prioritise when it comes to addressing the identified weaknesses in the model. The higher the number of sticky dots placed against an intervention would indicate its importance. The project team will collate the findings of the voting exercise following the workshop and the results will be presented in workshop 2. We have revised the relevant section (page 7) accordingly "The expert stakeholder group will work together to prioritize the interventions, policy initiatives or programs that they would like to see included in the model to capture their effects. This exercise will take place as a facilitated discussion and voting in line with Freebairn et al (2022)."

7) The parametrization of an SD model for suicide is a project by itself. There are plenty of unknowns in suicide (particularly at the community and societal levels), and navigating a map of suicide given these data limitations is tricky (see "The necessity and difficulty of navigating uncertainty to develop an individual-level computational model"). How are the authors planning to deal with the unknowns when operationalizing the model? 'Constrained optimization' is nice but if there is a lot of unknowns, there is a very large search space which brings both computational models and may lead to identifying values that are not actually possible.

Response: There is a wealth of good quality primary and secondary datasets to support research in the field of suicide prevention. Although data availability and quality may vary across regions and countries, there are commonly used strategies employed to address data limitations (Occhipinti et al, 2021). As explained on page 7, model parameterisation will draw on national, state, and local datasets (we have provided an indicative list of those datasets); empirical, peer-reviewed research (e.g., Cochrane systematic reviews); triangulation of multiple data sources; parameter estimation via constrained optimisation, and multidisciplinary expert and local knowledge to identify plausible estimates. In addition, as explained on page 8, sensitivity analyses will be performed to assess the impact of uncertainty in estimates of the direct effects of each intervention on the simulation results.

8) The workshops are intended to look at the complex situations surrounding people's lives in a local setting. However, every dataset mentioned in this project is at the national level. There seems to be a disconnect between the local workshops and the datasets available. Can the authors provide a table about which datasets would be used and for which domain (e.g., local healthcare resources, family dynamics, substance use and abuse...)?

Response: The list of datasets provided in the manuscript is indicative. We have revised this section to include additional examples of local datasets that will be used for model parameterisation (page 7). However, as the development of the computational model is ongoing, we are unable, at this stage, to provide a definitive list of secondary data sources that will be used. Local/regional data can be extracted from national datasets. For example, for the population sector of the model, data on births, deaths and migration for North-West Melbourne can be extracted from the Australian Bureau of Statistics. Data on mental health hospitalisations, suicide attempts/deaths in Birmingham can be extracted using the Clinical Record Interactive Search (CRIS) system.

9) Verification and validation are extremely important in a model, and they are two distinct activities. I could not find how verification would be carried out in the descriptions. As for validation, we expect the discrepancy between simulated outcomes and expected outcomes to be within an acceptable range, for example. It is fine to have some 'extreme testing' by running a few scenarios and asking

stakeholders that the projections are 'plausible', but it does not replace the need for comparisons of point estimates. Can the authors clarify what their model's outputs will be compared against, and what would be an acceptable accuracy?

Response: We have now revised the section on model validation (page 7-8) to provide further information about how verification and validation will be carried out.

The model will be validated by (i) testing whether the model could replicate historic time series data across a range of key indicators (self-harm hospitalisations, emergency department (ED) presentations, suicide deaths). Outputs will be calculated every 0.875 days (the numerical integration time step, dt , will be set to one-eighth of a week; Sterman, 2000) over a period of 20 years, starting from 2013, permitting comparisons of model outputs with historic data from 2013 to 2023, and forecasts of the impacts of intervention/policy scenarios simulated from the time of implementation (2023) to 2033; (ii) ensuring face validity of the model structure and performance by consulting our expert stakeholder group about whether the model and/or its behaviour are reasonable. During the participatory model building process, stakeholders will interact directly with the model interface to provide feedback on its design and functionality, run scenarios, test alternative assumptions, discuss results.

10) The proposal is interesting as it deals with resources, but that aspect seems to be missing from the process. For example, to find which combination of interventions have the most significant reductions, the answer could just be: all of them. But that is not a very useful answer, because we do not have the resources to perform every single intervention. Healthcare constraints or a cost-effectiveness aspects are important to guide intervention design and planning, otherwise the model can only tell us about the expected effect of an intervention but cannot be used to prioritize them. On the design of simulation tools for youth suicide prevention, it may be useful to look at "Design and Deployment of a Simulation Platform: Case Study of an Agent-Based Model for Youth Suicide Prevention".

Response: When simulating the impacts of alternative interventions or policies to determine what works (and what does not work), factors such as capacity, access, and workforce will be taken into consideration (page 7). These factors will help inform not only which interventions we should be investing in (or disinvesting) but also for how long, with what intensity, and how much we should be investing to gain a better understanding of how the impacts of programs introduced into that system might play out over time (Atkinson et al, 2020; Occhipinti et al, 2021). The advantage of such modelling is that local decision makers can explore what combination of interventions, at what time, in what sequence, will be most effective given the limitations of financial and human resources in the region. So even if the model indicated 'all of them' should be implemented (this is not often a finding), local stakeholders can see which combination is both feasible and impactful within current resources and also, which would be the next most impactful intervention to add to the portfolio if new funding could be additionally leveraged.

11) Translating from Australia to the UK is a very interesting final work package. Reading that the "conceptual model will be revised to reflect pathways and service provision relevant", I wonder whether this has ever been done. As in, is there prior work on creating a systems map with one group of participants and then asking another group to adapt it? My concern here is about confirmation bias. If I show a suicide map to a set of stakeholders, there would be a risk that they endorse most of its content, but they would not have produced it themselves if we ran the workshop from scratch.

Response: This is the first comparative case study using complex systems modelling and simulation for suicide prevention taking place in Australia and the UK that we are aware of. The models developed will be regional and will reflect regional population needs and service landscape. The health system in Australia is significantly different from the UK health system. The North-West

Melbourne conceptual model will be shared with the UK stakeholders for reference and to illustrate the process of conceptual model development. Although there will be aspects of the model which might be conceptually similar (e.g., how the population increases (via births/immigration) and decreases (via deaths/emigration), the needs and experiences of stakeholders as well as young people’s help-seeking journey and community/mental health service pathways will be different between the two regions.

12) I am glad to see that "young people advised on the participatory study design (workshops); ethical considerations and safety issues." However, I believe that sharing these details would be important for the protocol. There are definitively risks inherent to having conversations with young people who have a history of self-harm behaviors, and the management of these risks throughout the workshop is critical.

Response: Further information about the study’s Risk Management Strategy informed by young people from the Institute of Mental Health, University of Birmingham youth advisory group and Orygen’s Youth Council is now provided in Supplementary Material 3.

References

Atkinson JA, Skinner A, Lawson K, Rosenberg S, Hickie IB. (2020). Bringing new tools, a regional focus, resource-sensitivity, local engagement and necessary discipline to mental health policy and planning. *BMC Public Health*, 20:814.

Freebairn L, Atkinson J, Kelly P. Simulation modelling as a tool for knowledge mobilisation in health policy settings: a case study protocol. *Health Res Policy Syst* 2016;21, 14(1):71

Freebairn L, Occhipinti J, Huntley S. Participatory methods for systems modelling of youth mental health: An implementation protocol. *JIMR Res Protoc* 2022; 7, 11 (2): e32988

Giabbanelli, P.J., Rice, K.L., Galgoczy, M.C. et al. Pathways to suicide or collections of vicious cycles? Understanding the complexity of suicide through causal mapping. *Soc. Netw. Anal. Min.* 12, 60 (2022).

Occhipinti JA, Skinner A, Iorfino F, et al. Reducing youth suicide: systems modelling and simulation to guide targeted investments across the determinants. *BMC Med* 2021; 19(1):61 doi: 10.1186/s12916-021-01935-4

Sterman, JD. *Business Dynamics: Systems Thinking and Modeling for a Complex World*. McGraw-Hill, 2000.

van Vliet, M., Kok, K., & Veldkamp, A. (2010). Linking stakeholders and modellers in scenario studies: The use of Fuzzy Cognitive Maps as a communication and learning tool. *Futures*, 42(1), 1-14.

Voinov A., Kolagani N., McCall M.K., Glynn P.D., Kragt M.E., Ostermann F.O., Pierce S.A., Ramu P. (2016). Modelling with stakeholders - Next generation. *Environ. Model. Softw.* 77, C, 196–220.

VERSION 2 – REVIEW

REVIEWER	Giabbanelli, Philippe J. Miami University
REVIEW RETURNED	23-Jun-2023
GENERAL COMMENTS	I appreciate the thoughtful and detailed answers by the authors. The response letter has a trove of information. However, it appears at present that this information is solely for the benefits of

	the reviewers, as the main document has not reflected these changes apart from very minor edits. While I enjoy this scientific discussion, my primary concern is to ensure that readers have access to the information. I would much prefer a response letter that is extremely brief and simply tells me where changes have been made in reaction to each question (e.g., page X, lines Y-Z). I would be pleased to review this manuscript once again once the manuscript has been edited in reaction to each of the recommendations.
--	---

VERSION 2 – AUTHOR RESPONSE